# Heterostructure Cu$_2$O/(001)TiO$_2$ Effected on Photocatalytic Degradation of Ammonia of Livestock Houses

**Shihua Pu [1,2], Hao Wang [1,2], Jiaming Zhu [3]** **, Lihua Li [4,\*], Dingbiao Long [1,2,\*]** **, Yue Jian [1] and Yaqiong Zeng [1]**

[1] Chongqing Academy of Animal Sciences, Chongqing 402460, China; pu88962@126.com (S.P.); wanghaocau@163.com (H.W.); jiany16@163.com (Y.J.); zengyaqionghai@163.com (Y.Z.)
[2] Scientific Observation and Experiment Station of Livestock Equipment Engineering in Southwest, Ministry of Agriculture, Chongqing 402460, China
[3] College of Geology and Environment, Xi'an University of Science and Technology, Xi'an 710054, China; xrd610@126.com (J.Z.)
[4] College of Mechanical and Electrical Engineering, Agricultural University of Hebei, Baoding 071001, China
**\*** Correspondence: lilh890@163.com (L.L.); longt853@126.com (D.L.)

**Abstract:** In this paper, a heterogeneous composite catalyst Cu$_2$O/(001)TiO$_2$ was prepared by the impregnation-reduction method. The crystal form, highly active facet content, morphology, optical properties, and the photogenerated electron-hole recombination rate of the as-prepared catalysts were investigated. The performance of Cu$_2$O/(001)TiO$_2$ was appraised by photocatalytic degradation of ammonia under sunlight and was compared with lone P25, Cu$_2$O, and (001)TiO$_2$ at the same reaction conditions. The results showed that 80% of the ammonia concentration (120 $\pm$ 3 ppm) was removed by Cu$_2$O/(001)TiO$_2$, which was a higher degradation rate than that of pure P25 (12%), Cu$_2$O (12%), and (001)TiO$_2$ (15%) during 120 min of reaction time. The reason may be due to the compound's (Cu$_2$O/(001)TiO$_2$) highly active (001) facets content that increased by 8.2% and the band gap width decreasing by 1.02 eV. It was also found that the air flow impacts the photocatalytic degradation of ammonia. Therefore, learning how to maintain the degradation effect of Cu$_2$O/(001)TiO$_2$ with ammonia will be important in future practical applications.

**Keywords:** heterostructure; photocatalysis; ammonia; theoretical bases

## 1. Introduction

Ammonia is an important precursor of fine particulate matter PM2.5 in the atmosphere and promotes the formation of haze. China has been the one of largest ammonia emitters in the world [1]. An investigation of the NH$_3$ emission sources and contribution rate in 2006 shows that the total ammonia emissions of China are 980 t/a, more than 96% and 45% of Europe and the USA, respectively, and contributes to global and Asian emissions by approximately 15% and 35%, respectively; of which, livestock manure accounts for approximately 54% of the total emissions [2]. In terms of the global ammonia emissions, animal husbandry is clearly the major contributor for ammonia pollution, as it accounts for 70.9% of the total [3]. The ammonia from animal husbandry may not only cause global climate change, but can also bring disease in animals and humans, such as paralysis centralis, endobronchitis, and others [4]. Therefore, strengthening the control of ammonia pollution of animal husbandry has become an important task for agricultural, non-point source pollution treatment in China.

In the past, livestock farms adopted traditional physical methods of adsorption and ventilation to eliminate the ammonia generated in livestock and poultry houses [5]. Although these methods can decrease the release concentration of ammonia over time, they cannot reduce the total amount of ammonia emissions and may aggravate the secondary ammonia pollution of the environment. With more exacting livestock air quality requirements from public authorities, some large-scale livestock farms have tried to erect biological exhaust air cleaning facilities to reduce ammonia, dust, and odor emissions, so that they can get a license to build livestock houses close to small towns or near farming households. Biological filters can clean $NH_3$ emissions effectively, but limitations still exist, such as unstable operation and high replacement costs [6]; the end result is that most livestock farm ammonia emissions are not eliminated at all. Therefore, it is very important to develop a stable and efficient degradation material to deal with the air pollution from livestock farms.

As an ideal semiconductor photocatalytic material, titanium dioxide was discovered to be able to photodissociate water under ultraviolet irradiation by Fujishima in 1972 [7]. Since then, it has become a promising photocatalyst because of its good chemical stability, safety, non-toxicity, strong redox ability, and so on, and has been widely used in environmental control [8–12], exploitation of energy resources [13,14], bioengineering [15], self-cleaning material [16], anti-microbial [17], sensor [18], and other fields. In recent years, $TiO_2$ was used as pigment coating to reduce ammonia emissions under ultraviolet light [19,20], with a 90% conversion efficiency of 200 ppm $NH_3$ converted into $N_2$ at high temperature (673 K) [21]. In the reaction process, the degradation product of $NH_3$ is mainly $N_2$, with $N_2O$, $NO_2^-$, and $NO_3^-$ by-products [22,23]. However, $TiO_2$ can only respond to high energy ultraviolet light due to its wide band gap (3.2 eV) [24], which makes the utilization ratio of $TiO_2$ very low under sunlight with low ultraviolet content in the solar spectrum [25]. Meanwhile, the high photogenerated electron-hole recombination rate of $TiO_2$ leads to its catalytic activity [26]. All of these considerations contribute to serious obstacles to $TiO_2$ application in practice. To overcome these shortcomings, much research has been attempted to improve the performance of $TiO_2$ by adjusting the exposure ratio of its highly active surface or combining the material with other semiconductors. Adjusting the (001) facets ratio of $TiO_2$ (anatase) has been reported to be a efficiency way to improve the photocatalytic efficiency (Keyue Wu, [27]), but most of the available anatase crystals are primary composed of thermodynamic stable (101) surfaces [28], whose surface energy (0.44 J/m$^2$) is much smaller than the (001) surface (0.90 J/m$^2$) [29]. Another helpful way to facilitate charge separation and improve degradation performance is coupling with other semiconductors ($C_3N_4$ [30], PbS [31], $WO_3$ [32], $Cu_2O$ [33], CdS [34], and others) with small band gaps; this can not only enhance the spectral response, but also transfer photogenerated electrons from one reporter to another. Among these semiconductors with small band gaps, the oxides of copper ($Cu_xO$) are fascinating materials due to their remarkable optical, electrical, thermal, and magnetic properties [35], among the $Cu_xO$, $Cu_2O$ has the advantages of simple preparation processes, low raw material price, environmental friendliness, and a good response in the visible range. Compared with other transition metal oxides, $Cu_2O$ is a promising green catalyst for environmental protection. As a p-type direct semiconductor [36], $Cu_2O$ can be coupled with n-type semiconductor $TiO_2$ into a heterostructure. This heterostructure displays much better performance than the single material and is widely used for decomposing water [37], solar cells [38], and degrading pollutants [39]. At the same time, some interesting materials also need our attention. Syed et al. [40] synthesized $\alpha$-$Ga_2O_3$ by a sonication-assisted method, which has excellent photocatalytic activity is observed under solar light irradiation with an energy bandgap reduction, due to the presence of trap states, which are located at about 1.65 eV under the conduction band minimum. Datta et al. [41] synthesized 2D $\alpha$-$MoO_{3-x}$ nanosheets structural with defects and oxygen vacancies in the planar construction of molybdenum oxide nanosheets significantly increase the active sites of the catalyst, which act as key factors to promote the hydrogen evolution reaction (HER) performance.

In this paper, a heterogeneous composite catalyst $Cu_2O$/(001)$TiO_2$ was prepared by the impregnation-reduction method and was used to degrade ammonia in a livestock house under

sunlight for the first time, so as to realize the efficient utilization of solar energy and offer an efficient degradation material for the field of animal husbandry.

## 2. Results and Discussion

### 2.1. Morphology Analysis of Carrier (Polyester Fiber)

Figure 1 is a scanning electron microscope (SEM) view of the carrier (polyester fiber). It can be seen from the diagram that the carrier is folded by a lot of filaments into an interlaced three-dimensional structure, which can provide a large number of adhesion points to the catalysts.

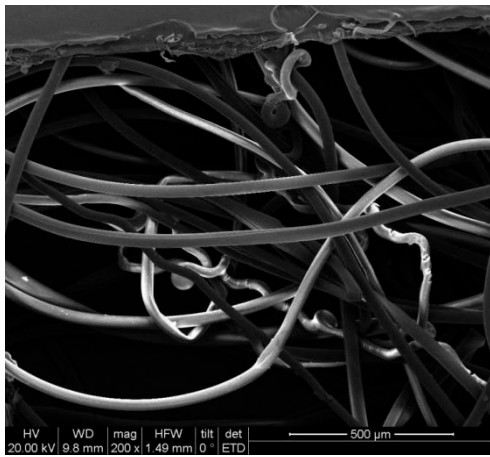

**Figure 1.** SEM image of the carrier.

### 2.2. X-ray Diffraction of Catalysts

Figure 2a showed the characteristic peaks of the prepared (001) $TiO_2$ catalyst at $2\theta$ = 25.28, 37.80, 48.04, 53.89, 55.06, 62.68, 70.31, 75.03 corresponding to (101), (004), (200), (105), (211), (204), (220), and (215) crystal facets, respectively, almost the same as the anatase $TiO_2$ (JCPDS no.21-1272) standard card [42]. The characteristic peaks of $Cu_2O$ appeared at $2\theta$ = 29.63, 36.50, 42.40, 52.58, 61.52, 73.70, and 77.57, corresponding to (110), (111), (200), (211), (220), (311), and (222) crystal facets, almost in accordance with the standard card of $Cu_2O$ (JCPDS no.65-3288) [43]. The composite catalyst $Cu_2O/(001)TiO_2$ showed obvious characteristic peaks at $2\theta$ = 25.28, 37.80, 48.04, 55.06, 62.68, corresponding to (101), (004), (200), (211), and (204) crystal facets, respectively, basically consistent with anatase $TiO_2$. The composite catalyst $Cu_2O/(001)TiO_2$ showed obvious characteristic peaks at $2\theta$ = 29.63, 42.40, 52.58, 61.52, 73.70, 77.57, corresponding to (110), (200), (211), (220), (311), and (222) crystal facets, almost in accordance with the standard card of $Cu_2O$. The composite catalyst $Cu_2O/(001)TiO_2$ showed weak characteristic peaks at $2\theta$ = 32.62, 38.73, 48.82, corresponding to (110), (111), (202) crystal facets, respectively, almost in accordance with the standard card of CuO (JCPDS no.89-5895). The composite catalyst $Cu_2O/(001)TiO_2$ showed the characteristic peak around at $2\theta$ = 43.29, maybe corresponding to (111) crystal facet of Cu (JCPDS no.04-0836). From Figure 2b, it can be seen that there was no S element in the $Cu_2O/(001)TiO_2$ composite, which indicated that the reaction was sufficient and $CuSO_4$ was not contained in the product. In summary, the prepared composite catalyst $Cu_2O/(001)TiO_2$ contains CuO and may also contain Cu.

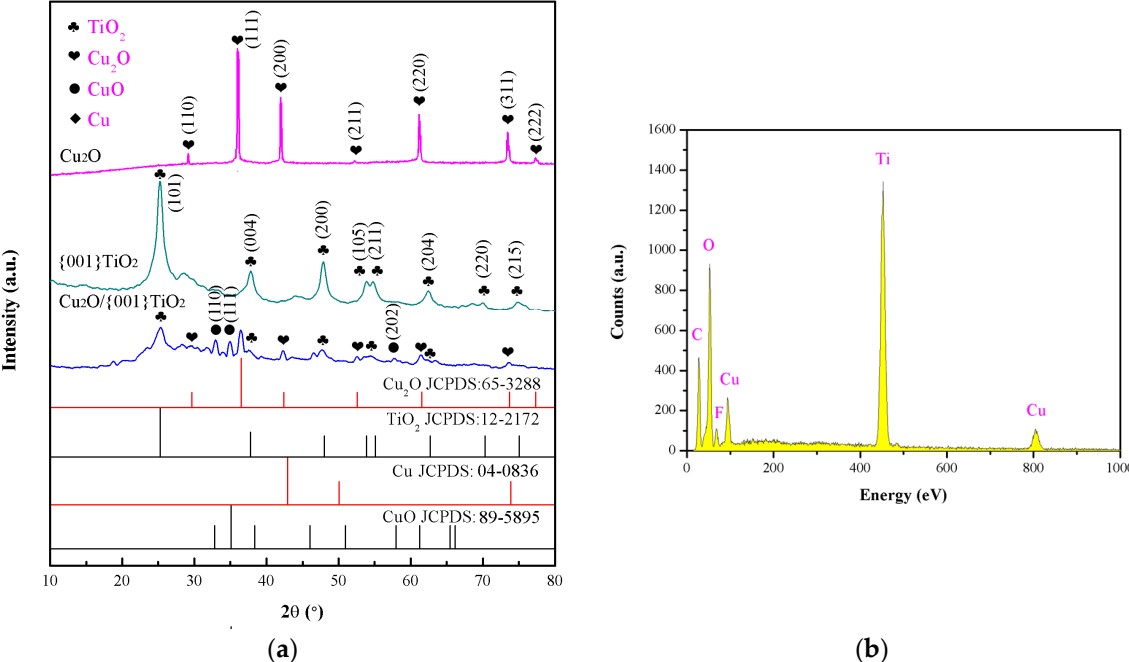

**Figure 2.** (**a**) XRD patterns of Cu$_2$O, TiO$_2$, and Cu$_2$O/(001)TiO$_2$; (**b**) EDS patterns of Cu$_2$O/(001)TiO$_2$.

### 2.3. Crystal Facet Analysis of Catalysts

Figure 3a contains spacings of d = 0.352 nm and d = 0.235 nm, corresponding to the (101) and (001) facets of TiO$_2$, respectively [44]. This shows that the prepared TiO$_2$ has the high activity (001) facets. The Figure 3b shows d = 0.235 nm and d = 0.245 nm spacings, corresponding to the (001) facets of TiO$_2$ and the (111) surface of Cu$_2$O, respectively [45], which shows that the prepared composite catalyst contains the high activity (001) facets.

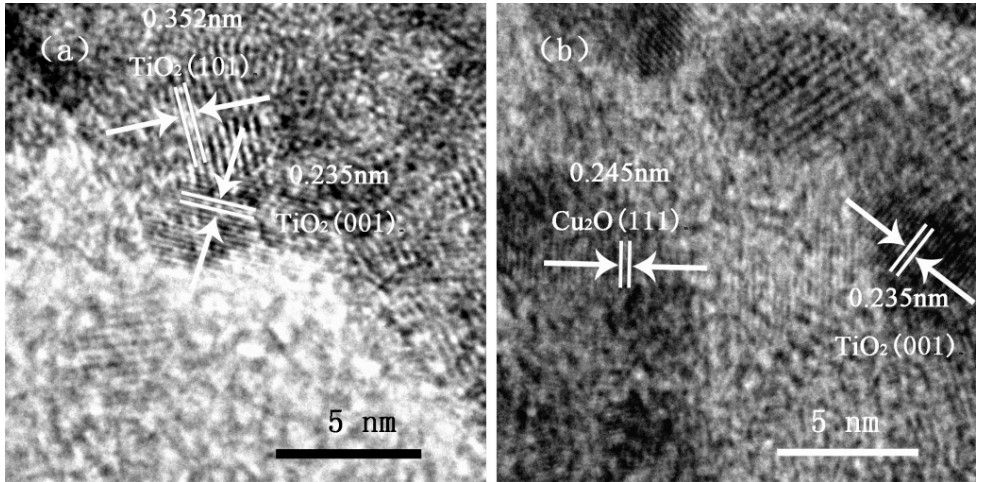

**Figure 3.** HRTEM images of the catalysts. (**a**) (001)TiO$_2$; (**b**) Cu$_2$O/(001) TiO$_2$.

### 2.4. Raman Analysis of Catalysts

It can be seen in Figure 4. The peak at 144 cm$^{-1}$ was marked as Eg and 514 cm$^{-1}$ was marked as A1g. The ratio of peak strength at A1g to peak strength at Eg is the (001) exposure ratio of crystal facets. The calculated (001) facet content of prepared TiO$_2$ is I(101)/I(004) = 29% [46], which is significantly lower than that of the compounded, calculated to be I(101)/I(004) = 37%. The reasons for this may be as follows:

$$CuSO_4 + 2NaOH = Cu(OH)_2 \downarrow + Na_2SO_4 \tag{1}$$

$$2Cu(OH)_2 + C_5H_{11}O_5{-}CHO \rightarrow C_5H_{11}O_5{-}COOH + Cu_2O \downarrow + 2H_2O \qquad (2)$$

In the experiment, the amounts of $CuSO_4$ and NaOH were 4.99 g and 4 g, respectively. Through the calculation of Reaction (1), it was found that the amount of NaOH is excessive. Assuming that all excessive sodium hydroxide existed in the whole system, the concentration range is about 0.197–0.317 mol/L. In the process of adding glucose drop by drop, the concentration of NaOH was also decreasing gradually. It was found that NaOH and $(001)TiO_2$ not only increased the content of (001) facets, but also changed the morphology of the catalysts, which was consistent with the results of Hou et al. [47].

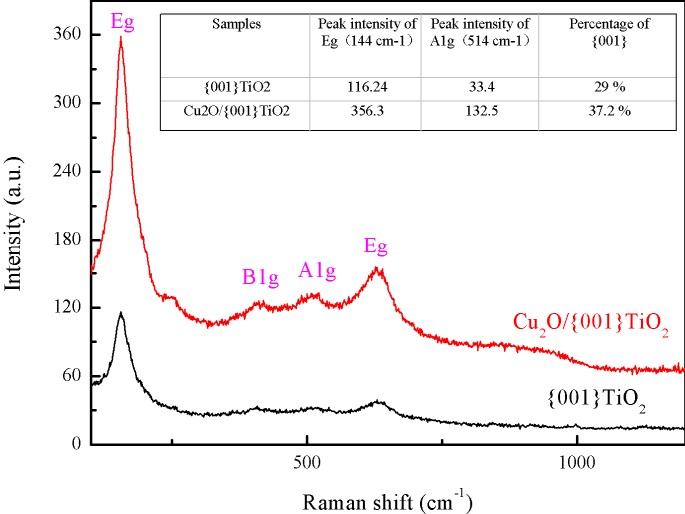

| Samples | Peak intensity of Eg (144 cm-1) | Peak intensity of A1g (514 cm-1) | Percentage of {001} |
|---|---|---|---|
| {001}TiO2 | 116.24 | 33.4 | 29 % |
| Cu2O/{001}TiO2 | 356.3 | 132.5 | 37.2 % |

**Figure 4.** Raman spectrum of the catalysts.

### 2.5. Catalyst Morphology Analysis

Figure 5 shows the SEM morphology images of the catalysts. From Figure 5a, it can be seen that the prepared $Cu_2O$ crystals have no obvious agglomeration phenomenon and the size is basically uniform. Figure 5b reveals a smooth, spherical surface; Figure 5c displays a slight agglomeration of $(001)TiO_2$, which forms an irregular spherical shape by extrusion. Figure 5d exhibits a flat sphere with concave and convex surfaces; Figure 5e,f shows that the prepared $Cu_2O/(001)TiO_2$ has uniform dispersion and different sizes; the composite $(001)TiO_2$ was no longer a crushed sphere with a rough surface, but a crushed sphere. This is due to the influence of residual sodium hydroxide. Raman analysis in this experiment showed that excess sodium hydroxide was present, which indicates that the conjecture is correct. The composite $Cu_2O$ was no longer globular, but flaky. This may be due to the influence of $(001)TiO_2$ on the $Cu^{+1}$ crystal nucleus adhering to the $(001)TiO_2$ surface during in situ growth, which cannot grow in the same way as before. This resulted in the growth of new pancake-like catalysts; the new morphology can provide a better place for the separation of photogenerated carriers.

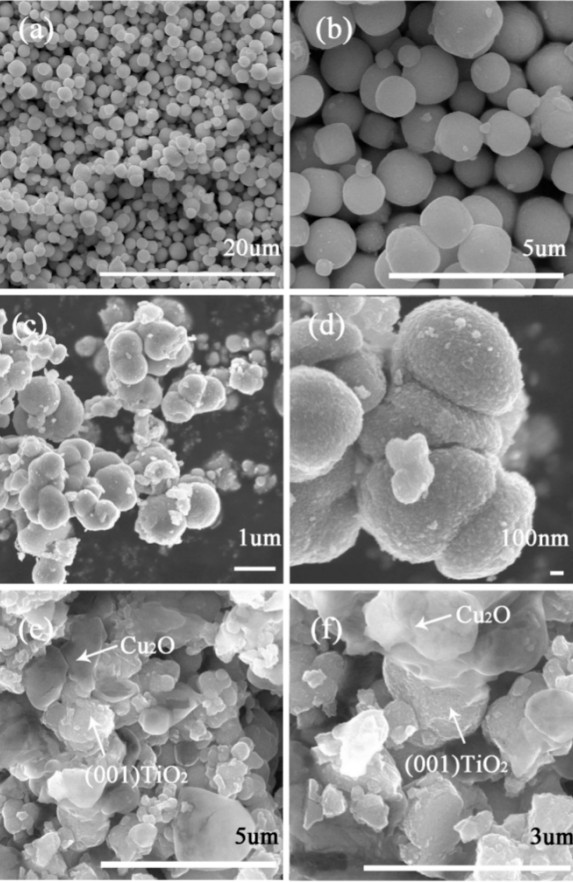

**Figure 5.** SEM images of the catalysts. (**a,b**) Cu$_2$O; (**c,d**) (001)TiO$_2$; (**e,f**) Cu$_2$O/(001)TiO$_2$.

## 2.6. Optical Characterization and Band Gap Energy

Figure 6a showed the UV-Vis absorption spectra of the catalysts. We found that the absorption of (001)TiO$_2$ is slightly stronger than that of Cu$_2$O/(001)TiO$_2$ in the range of 200–328 nm, while in the range of 328–800 nm, the absorption enhancement of the composite catalyst Cu$_2$O/(001)TiO$_2$ is much stronger than that of the (001)TiO$_2$, which indicated that Cu$_2$O/(001)TiO$_2$ has strong light absorption ability mainly in the visible range. The Diffuse Reflectance Spectra (DRS) of the catalysts were measured and shown in Figure 6b, which is useful to study the optical properties of the materials and the band gap. From the DRS spectra is possible the determination of Eg by applying the Kubelka–Munk method using the following equations (Equations (3) and (4)) [48]:

$$F_{(R)} = (1 - R)^2/2R \tag{3}$$

$$(hvF) \sim (hv - Eg)^2 \tag{4}$$

where, F is the Kubelka–Munk function, R is the reflectance, hv is the photon energy, and Eg is the band gap. By plotting $(hvF)^{1/2}$ for indirect allowed transitions versus hv, the Eg of the semiconductor samples can be obtained. The energy bang gap of the different samples can be calculated by the linear fit of the curve reported in the Figure 6c and the Eg values are determined by the intercept in the x-axis. Reported in Table 1 are the band gap energy values of the different samples calculated with the Kubelka–Munk method.

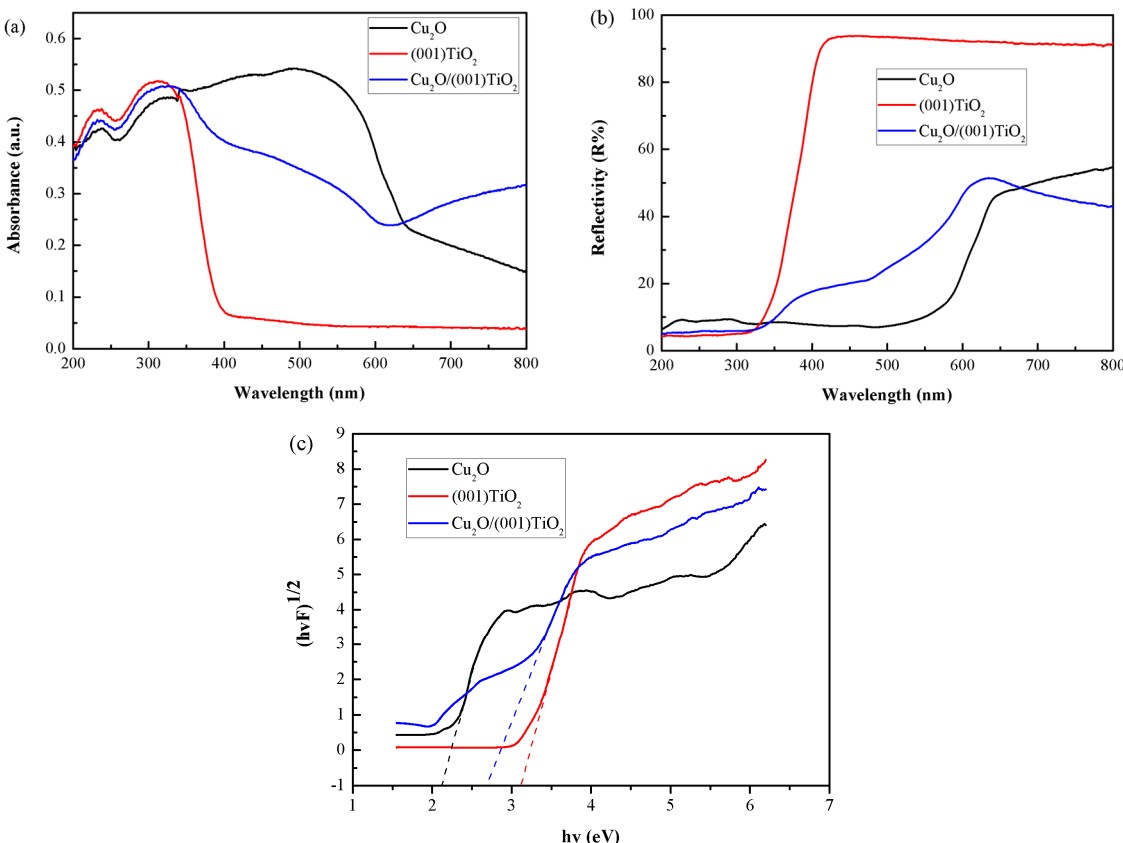

**Figure 6.** (**a**) UV-vis Light Diffuse Reflectance spectra of the catalysts; (**b**) DRS spectra of the samples; (**c**) Band gap calculation from DRS spectra by Kubelka–Munk method of the materials.

**Table 1.** The band gap energy values of the different samples.

| Samples | Eg (eV) |
|---|---|
| $Cu_2O$ | 2.12 |
| $(001)TiO_2$ | 2.68 |
| $Cu_2O/(001)TiO_2$ | 3.14 |

The band gaps of $Cu_2O$, $(001)TiO_2$, and $Cu_2O/(001)TiO_2$ are 2.12, 2.68, and 3.14 eV, respectively, which indicates that the composite $Cu_2O/(001)TiO_2$ is more easily excited than the single $(001)TiO_2$, which enhanced its activity.

### 2.7. XPS Analysis of Catalyst

The elements and valence states of $Cu_2O/(001)TiO_2$ were analyzed by XPS, as shown in Figure 7a. A survey spectra of $Cu_2O/(001)$ $TiO_2$ is seen, which contains information on C 1s, Ti 2p, O 1s, F 1s, and Cu 2p. Figure 7b showed that the binding energies for Ti 2p 3/2 and Ti 2p 1/2 are 458.90 and 464.59 eV, respectively, because Ti existed in the structure as $Ti^{4+}$ [49,50], which was favorable for charge transfer between $TiO_2$ and $Cu_2O$. Figure 7c showed two characteristic peaks of O 1S in $Cu_2O/(001)TiO_2$, lattice oxygen of catalysts at 530.11 eV, and water molecules adsorbed on the catalyst surface at 532.64 eV [51,52]. Figure 7d showed peaks near 932.90 and 952.77 eV, which are characteristic of $Cu^{+1}$ [53,54].

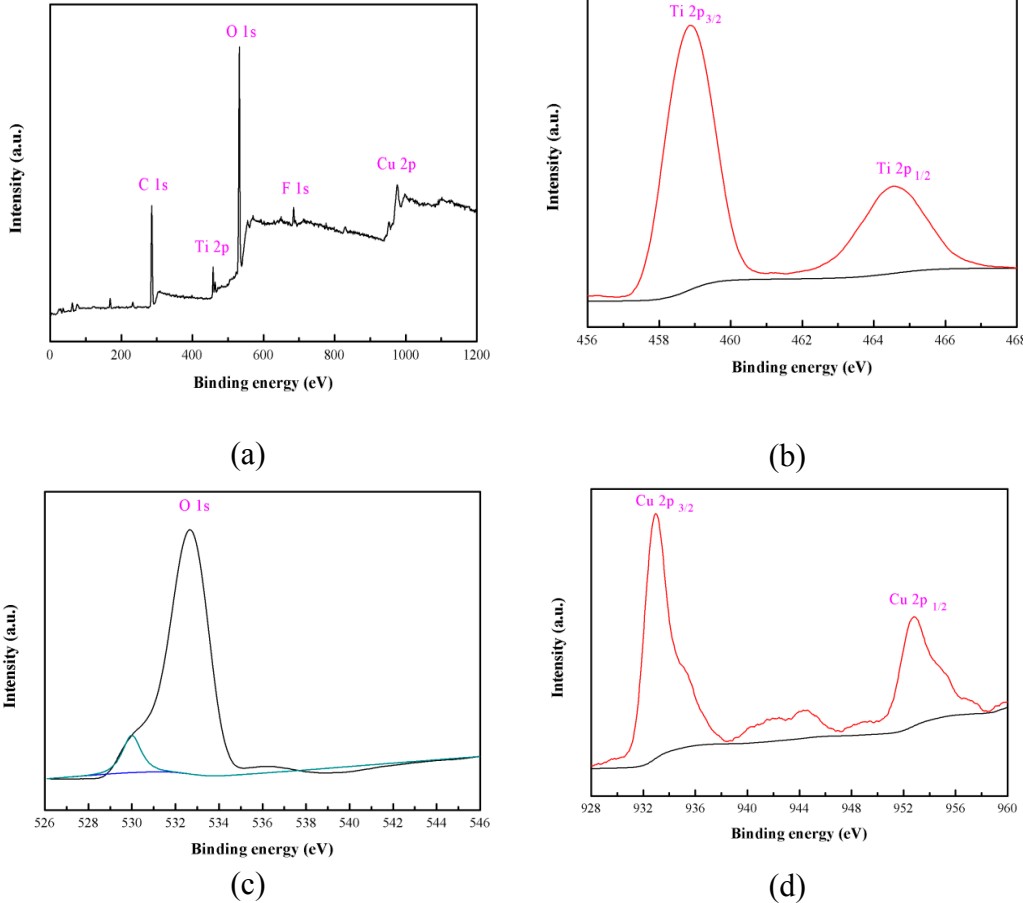

**Figure 7.** XPS spectra of the as-prepared $Cu_2O/(001)TiO_2$ sample: (**a**) survey spectrum, (**b**) Ti 2p spectrum, (**c**) O 1s spectrum and (**d**) Cu 2p spectrum.

### 2.8. Fluorescent Analysis of Catalysts

Fluorescence spectra were used to characterize the photogenerated electron-hole combination of the photocatalysts. The higher the photogenerated electron-hole binding law, the stronger the fluorescence intensity [55]. It can be seen from Figure 8 that the photogenerated electrons and photogenerated holes of $(001)TiO_2$ were greatly reduced after the introduction of $Cu_2O$ composite, which significantly improved the degradation efficiency of the catalyst during the degradation process [56].

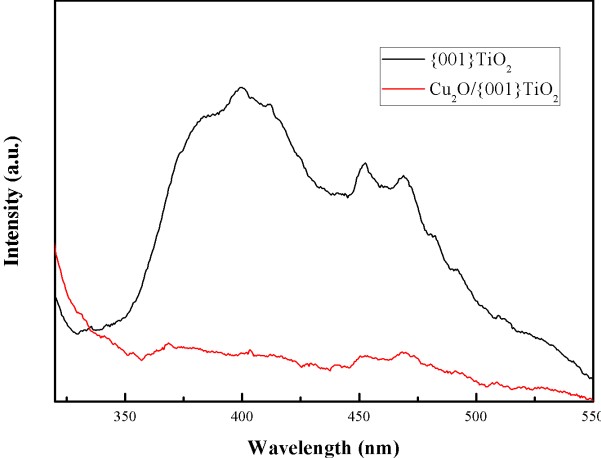

**Figure 8.** Fluorescence spectra of catalysts.

*2.9. Compound Mechanism of Cu₂O/(001)TiO₂*

We will explore the reasons for the improvement of photocatalytic performance by discussing the compound material mechanism. The conduction band position of semiconductor can be calculated by empirical formula [57]:

$$E_{CB} = \chi - Ec - Eg/2$$
$$E_{VB} = E_{CB} + Eg$$

where, $E_{CB}$ is the conduction band energy of semiconductors, $\chi$ is the geometric average of the absolute electronegativity of atoms in semiconductors, Ec is a constant relative to the standard hydrogen electrode (4.5 eV), Eg is the band gap width of semiconductors, and $E_{VB}$ is the valence band energy of semiconductors. The $\chi$ values of (001)TiO₂ and Cu₂O are about 5.86 and 5.33 eV, respectively.

The Eg values of (001)TiO₂ and Cu₂O were 3.14 and 2.12 eV, the $E_{CB}$ values of (001)TiO₂ and Cu₂O were −0.21 and −0.23 eV, and the $E_{VB}$ values were 2.93 and 1 eV, respectively, as shown in Figure 9a. Therefore, the electron transfer took place from the p-type semiconductor Cu₂O onto the surface of n-type semiconductor (001)TiO₂, which is helpful for the separation of light carriers and the catalytic performance of the composite catalyst, since $E_F$ of Cu₂O and (001)TiO₂ have differences in energy before and after catalyst contact. When p-type semiconductor contacts with n-type semiconductor, since the carriers type and concentration are different on both sides of the contact surface, the holes in the p-region are diffused toward the n-region, and the electrons in the n-region are also diffused into the p-region. This results in a decrease in the hole concentration on the side of the p-region near the interface, and a decrease in the electron concentration on the side of the n-region near the interface, so that there are almost no carriers that can move at the interface. Therefore, a positive space charge is generated in the n-region, and a negative space charge is generated in the p-region. These space charges form a self-built electric field near the interface, and the electric field direction points from the n-region to the p-region. Under the action of this electric field, under the action of this electric field, the carrier will drift and its direction is exactly opposite to the diffusion flow. The self-built electric field makes part of the holes entered the n-region return to the p-region, part of the electrons entered the p-region return to the n-region, and finally the diffusion flow and the drift flow reach an equilibrium state, and generally there is no macroscopic flow of carriers, as shown in Figure 9c. If there have no external electric field on either side of the p-n junction, then the entire system Fermi level should be the same. However, the existence of the space charge region will cause a self-built electric field in the vicinity of the interface, so that there is a potential difference near the p–n junction region, which eventually causes the energy band to bend, as shown in Figure 9b [58].

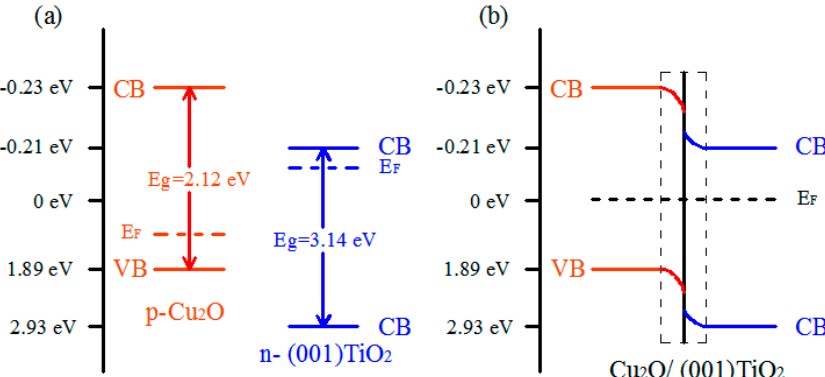

**Figure 9.** *Cont.*

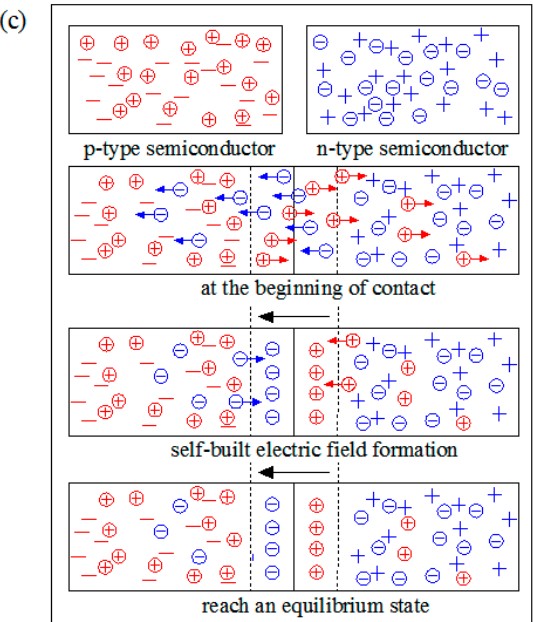

**Figure 9.** Energy band diagrams for (**a**) $Cu_2O$ and $TiO_2$ before contact; (**b**) $Cu_2O/TiO_2$ composite; (**c**) Charge variations before and after contact of different types of semiconductors.

### 2.10. Degradation of Ammonia by the Catalysts

### 2.10.1. Controlled Test of Catalyst Carrier (Polyester Fiber Carrier)

The control experiment was conducted on the blank polyester fiber carrier (PET) that was not loaded with any photocatalyst; the absorptive photocatalytic efficiency of the blank PET on ammonia was measured in the presence or absence of light (Figure 10) within 120 min. Under the conditions of both dark and light, the degradation efficiency of the PET carrier on ammonia were 8% ± 1% respectively and the efficiency is not obviously improved, which means that the carrier of the PET itself hardly occurs photolysis. At the same time, the degradation rate of light is slightly lower than that of no light. It is possible that the temperature produced by light affects the stability of the gas.

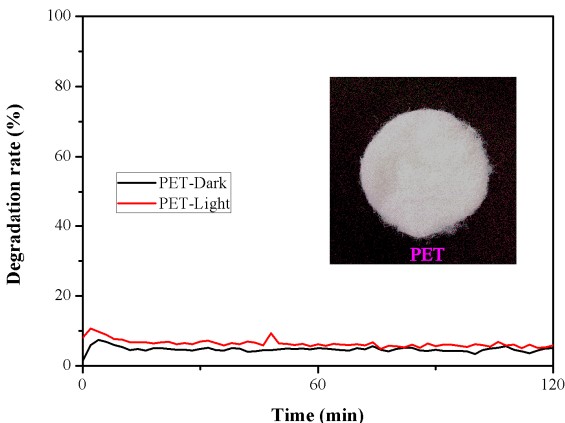

**Figure 10.** Controlled test of catalyst carrier under light and dark conditions.

### 2.10.2. Degradation of Ammonia by Different Catalysts

As can be seen from Figure 11, the degradation rates of ammonia by single P25, $Cu_2O$, and (001)$TiO_2$ continued to decline in the first 30 min until about 15% ± 2%. This may be due to the fact that the sunlight itself is not pure ultraviolet light, so that the number of active sites can be activated may be relatively less. This means that many ammonia molecules were brought to the tail gas

treatment area without contact with the catalyst. The degradation rate of ammonia by the composite catalyst $Cu_2O/(001)TiO_2$ maintained above 80% in the first 90 min, but decreased significantly in 90–120 min. This may be due to photocorrosion. Overall, the degradation effect of the composite catalyst was better than that of the single catalysts P25, $Cu_2O$, and $(001)TiO_2$.

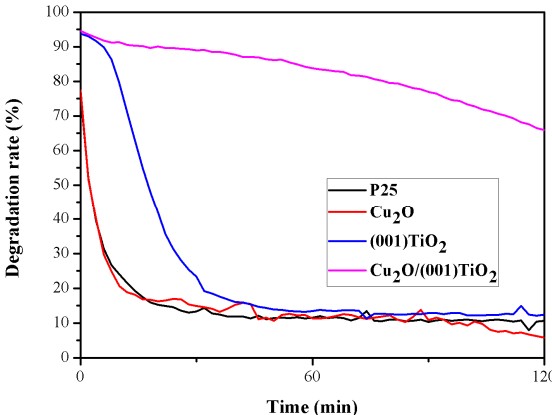

**Figure 11.** Degradation of ammonia by different catalysts.

### 2.10.3. Degradation of Ammonia by Air Flow Rate

In the process of photocatalytic reaction of ammonia, the final reaction is carried out on the surface of photocatalyst, which generally undergoes the following continuous processes: ammonia diffuses to the surface of photocatalytic material; ammonia diffuses from the outer surface to the inner surface of the catalyst; ammonia molecules are adsorbed on the catalyst material; the adsorbed ammonia undergoes photocatalytic reaction; the products produced after photocatalytic reaction are desorbed from the surface of the catalyst; the products diffuse from the surface of the material to the outside surface; and the products desorb from the outside surface of the material to the air.

As can be seen from Figure 12, the air flow rate was varied between 0.5 min/L, 1 min/L, and 2 min/L. In this experiment, when other initial conditions were the same and the gas flow rate was gradually increased, the probability of ammonia adsorbed by the material decreases in unit time, the residence time on the catalyst surface was shortened, only $NH_3$ adsorbed on the catalyst surface can be degraded, so the photocatalytic reaction was insufficient, and ultimately the degradation efficiency is reduced. Therefore, in the photocatalytic degradation of ammonia, it is very important to control the appropriate air flow rate.

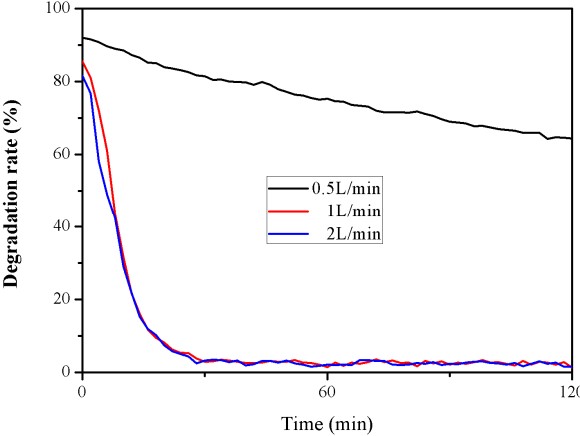

**Figure 12.** Effect of air velocity on ammonia degradation.

### 2.10.4. Catalyst Reutilization Performance

As can be seen from Figure 13, the degradation rate of ammonia can be maintained at about 60% after three times of use, but the degradation effect will become worse as the number of times of use increases. This was because there was a loss of catalyst for each repeated use.

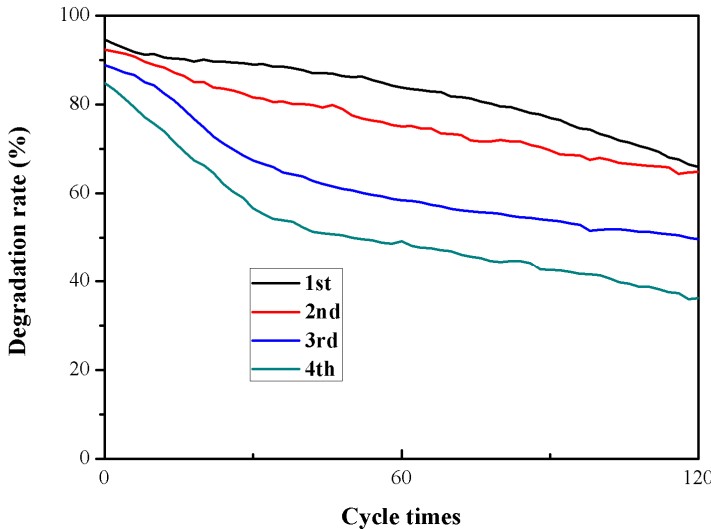

**Figure 13.** Reusing performance of catalyst.

## 3. Materials and Methods

### 3.1. Materials

Butyl titanate, anhydrous glucose, sodium hydroxide (AR, Chengdu Kelong Chemical Reagent Factory, Chengdu, China), anhydrous ethanol (AR, Chongqing East Sichuan Chemical Co., Ltd., Chongqing, China), hydrofluoric acid (AR, Sichuan Xilong Chemical Co., Ltd., Chongqing, China), anhydrous copper sulfate (AR, Tianjin Da Mao Chemical Reagent Factory, Tianjin, China) and P25 (Degussa, Qingdao, China).

### 3.2. Preparation of (001)TiO$_2$ Catalyst

Anhydrous ethanol (15.20 mL) was slowly added to 17.60 mL of butyl titanate and stirred until the solution was well distributed (designated solution A). Then, 15.20 mL of absolute ethanol was slowly added to 90 mL of ultra-pure water; 6 mL of hydrofluoric acid was then added to this solution (designated solution B). Next, the A solution was added to the B solution and was stirred under medium speed at room temperature for 2 h; the TiO$_2$ gel was allowed to sit for 2 days and then moved to a stainless-steel reactor containing a PTFE inner container at 100 °C for 2 h. After cooling, the precipitates were separated and washed alternately with ultra-pure water and absolute ethanol for three times, drying in a 100 °C drying oven. After completion of a grinding process, the obtained powder was marked as (001)TiO$_2$.

### 3.3. Preparation of Cu$_2$O/(001)TiO$_2$ Catalyst

CuSO$_4$ (4.99 g) was weighed and dispersed in 100 mL distilled water. Then, 12.48 g (001)TiO$_2$ was added to form a mixture. Ultrasound was used for 20 min. NaOH (4 g) was dissolved in 20 mL of distilled water; glucose (7.5 g) was dissolved in 75 mL of distilled water; then, the dissolved sodium hydroxide solution was added to the mixture of CuSO$_4$ and (001)TiO$_2$ dropwise, forming a blue precipitate. Then, the glucose solution was placed on a constant temperature magnetic stirrer and stirred continuously until the solution was heated to 34 °C. The heated glucose solution was added to the blue precipitation solution dropwise; then, the mixed solution was placed on a constant

temperature magnetic stirrer and was stirred continuously, heated, and reacted until the solution reached 70 °C, and kept for 15 min at that temperature. During the whole process, the solution gradually changed from blue to dark green. Eventually, a brick-red precipitate appeared. The brick-red precipitate solution was centrifuged and then washed repeatedly with deionized water for three times. The brick-red substance was obtained by drying at 60 °C. $Cu_2O/(001)TiO_2$ was obtained by grinding the brick-red substance into powder.

### 3.4. Catalyst Powder Loading

In order to prevent the powder catalyst from blowing away, polyester fiber cotton was used as a carrier; this material has good light transmittance, is light-weight, and is easy to operate. The cotton was treated with 5 mol/L NaOH for 2 h, was washed repeatedly with deionized water until its pH was neutral, and was dried in a blast drying oven at 60 °C.

Before the catalysts from Sections 2.2 and 2.3 were dried, the treated carriers were put into the washed catalysts solution, ran at a medium speed for 4 h in an oscillator, and were removed and dried in a blast drying chamber at 60 °C.

### 3.5. Catalyst Characterization

The crystal phase characteristics of the catalysts were measured by X-ray diffractometry (XRD, D8 Advance, Bruker, Rheinstetten, Germany) with Cu K$\alpha$radiation ($\lambda$ = 0.154 nm) under 40 kV at a scan speed of 6°/min. The optical properties of the catalysts were characterized with a UV-Vis spectrophotometer (UV-Vis, U-3010, Hitachi, Tokyo, Japan). The detailed lattice spacing of the as-synthesized samples was evaluated by a high-resolution transmission electron microscope (HRTEM, Tecnai G2 F20, FEI, Hillsboro, OR, USA). The crystal facet contents of the catalysts were analyzed by using Raman spectroscopy (Raman, LabRAM XploRA INV, Horiba, France). The morphology of the catalysts was observed by a field-emission scanning electron microscope (FE-SEM, Inspect F50, Thermo Fisher Scientific, Waltham, USA) with an accelerating voltage of 25 kV. The elemental composition and element valence state of samples were analyzed by X-ray photoelectron spectroscopy (XPS, EscaLab 250Xi, Thermo Fisher Scientific, Waltham, USA). The photoelectron hole recombination of samples was evaluated by Photoluminescence emission spectra (PL, F-2700, Hitachi, Tokyo, Japan).

### 3.6. Experimental Equipment of Photocatalytic Ammonia Gas

As shown in Figure 14. A small air pump with a measuring range of 0–3 L/min was used as the source of air. The air from the air pump became clean air after being treated with activated carbon and silica gel; this clean air sent ammonia vaporized from ammonia water to the photocatalytic reaction chamber. The sampling inlet and sampling outlet were set before and after the photocatalytic reaction chamber, respectively, which were connected to the inlet and outlet of an INNOVA continuous detector to reflect the changes of concentration in the whole photocatalytic process. In the photocatalytic reaction chamber, a 300 W xenon lamp was used to simulate sunlight and a condensing reflux tube was installed outside the light source to reduce the influence of a large amount of heat generated by the light source on the experimental results; the remaining tail gas passed through the tail gas treatment device. The photocatalytic reaction tube was made of a quartz tube with a length of 6 cm, an outer diameter of 3 cm, and an inner diameter of 2.7 cm. The average load of the catalysts was 0.18 g. The remaining tail gas was treated by the tail gas treatment device.

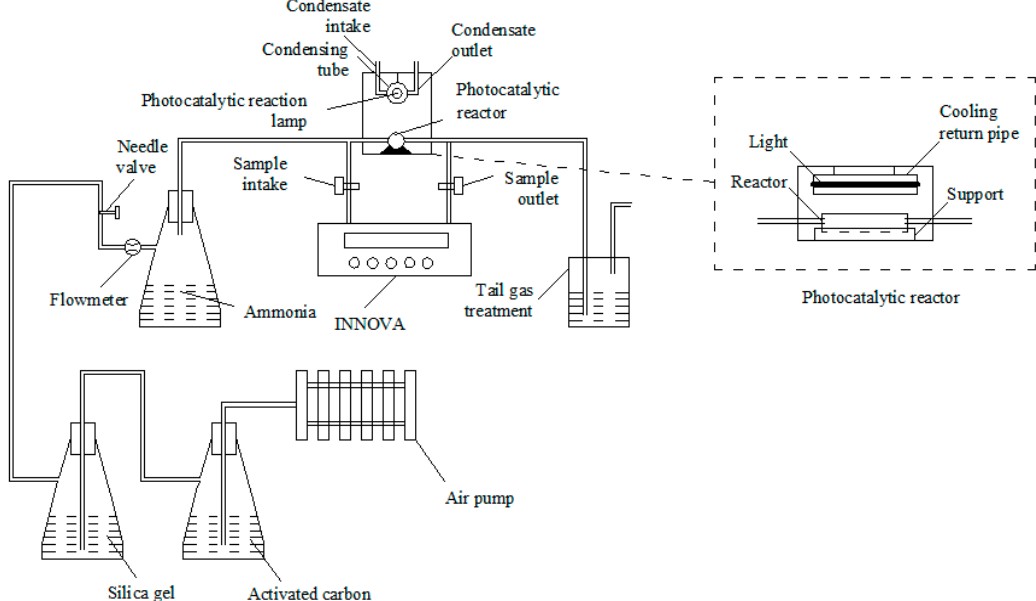

**Figure 14.** Schematic diagram of ammonia degradation unit.

There are two test ports before and after the reaction tube for ammonia degradation, namely, the sample intake and sample outtake, which are connected to INNOVA's intake and outtake, respectively. INNOVA is an instrument for continuously detecting ammonia concentration changes online. The degradation rate at this moment is as follows:

$$\text{The degradation rate (\%)} = \frac{C_{\text{in}} - C_{\text{out}}}{C_{\text{in}}} \times 100\%$$

where, $C_{\text{in}}$ is the ammonia concentration at the inlet and $C_{\text{out}}$ is the concentration of ammonia after the photocatalytic reaction.

*3.7. Photocatalytic Test*

3.7.1. Controlled Test of Catalyst Carrier (Polyester Fiber)

The blank polyester fiber without any catalyst was placed in the sealed reactor. The temperature of the reactor was about 18 °C. The ammonia concentration was 120 ± 3 ppm, the air flow rate was 0.5 L/min, and the 300 W xenon lamp was used as the light source for the photocatalytic reaction.

3.7.2. Degradation of Ammonia by Different Catalysts

The polyester fibers loaded with P25, $Cu_2O$, (001)$TiO_2$, $Cu_2O$/(001)$TiO_2$ were placed in the sealed reactor. The temperature of the reactor was about 18 °C. The ammonia concentration was 120 ± 3 ppm, the air flow rate was 0.5 L/min, and the 300 W xenon lamp was used as the light source for the photocatalytic reaction.

3.7.3. Degradation of Ammonia by Air Flow Rate

The polyester fibers loaded with $Cu_2O$/(001)$TiO_2$ was placed in the sealed reactor. The temperature of the reactor was about 18 °C. The ammonia concentration was 120 ± 3 ppm, the air flow rate were 0.5, 1, 2 L/min, separately, and the 300 W xenon lamp was used as the light source for the photocatalytic reaction.

### 3.7.4. Catalyst Reutilization Performance

The polyester fibers loaded with $Cu_2O/(001)TiO_2$ was placed in the sealed reactor. The temperature of the reactor was about 18 °C. The ammonia concentration was $120 \pm 3$ ppm, the air flow rate was 0.5 L/min, and the 300 W xenon lamp was used as the light source for the photocatalytic reaction, after the reaction, the carrier was removed and baked at 70 °C for 30 min for the next reaction.

### 4. Conclusions

In this paper, the preparation of $(001)TiO_2$ by the sol-gel method has been achieved and a heterogeneous structure of $Cu_2O/(001)TiO_2$ has been prepared by the impregnation-reduction method. XRD, SEM, PL, and other characterizations can confirm that $Cu_2O$ and $(001)TiO_2$ were successfully compounded together. XPS can confirm that Cu was present in the form of $Cu^{+1}$. The composite catalyst not only absorbed strongly in the visible light range, but also can be easily excited because of its small band gap, which made up for the shortcoming of low utilization of solar light by single $TiO_2$. The results showed that the effect of $Cu_2O/(001)TiO_2$ composite catalyst on ammonia degradation is obviously better than that of single P25, $Cu_2O$, and $(001)TiO_2$. We also found that air flow has a very important impact in the whole photocatalytic ammonia gas. However, a great deal of dust in the colonies, preventing catalyst loss, and light corrosion are challenges that were present in this work.

**Author Contributions:** In this paper, S.P., L.L. and D.L. designed the experiments; H.W. conducted the experiments; J.Z. analyzed the data; Y.J. and Y.Z. analyzed the characterized the results. S.P. wrote the article.

**Funding:** This work was funded by the National Key R&D Program of China (2018YFD0800100), Chongqing Financial Special Funds Project-International Cooperation (19502), the earmarked fund for Modern Agro-industry Technology Research System (CARS-35), the agricultural development finance program of Chongqing (16406) and Scientific Observation and Experiment Station of Livestock Equipment Engineering in Southwest of China, Ministry of Agriculture and Rural Affairs.

**Acknowledgments:** We would like to thank LetPub (www.letpub.com) for its linguistic assistance during the preparation of this manuscript.

**Conflicts of Interest:** There is no conflict of interests exiting in the manuscript submission, and it is approved by all of the authors for publication. All the authors listed have approved the manuscript to be enclosed.

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
