# Peer review of "Heterostructure Cu2O/(001)TiO2 Effected on Photocatalytic Degradation of Ammonia of Livestock Houses"

_catalysts, doi:10.3390/catal9030267_

Round 1
Reviewer 1 Report
The authors presented a composite Cu2O/TiO2 photocatalyst for ammonia degradation. Major flaws in the experimental design and reporting make it impossible for me to evaluate the performance of the catalyst or the merit of this work. I will need to see a revised version before I can provide a recommendation. Specific issues include:
Typically, photodegradation experiments report the change of the concentration of the pollutant that is being degraded. Here, instead of concentration, a "degradation rate" is reported. No definition of what the authors mean by degradation rate is provided in the manuscript. For instance, what does a degradation rate of 95% mean? Why does the degradation rate drop to 15%? How is the degradation rate calculated? How does the "INNOVA continuous detector" characterize the degradation - is it a gas chromatograph? All these questions need to be answered.
A doubling of gas flow from 0.5L/min to 1L/min caused a dramatic change in the photodegradation profile. This does not make sense and needs to be properly explained. The authors' explanation is inadequate.
Although the XRD spectrum of the Cu2O/TiO2 composite contains peaks from both TiO2 and Cu2O, many other peaks can be seen. It is likely that other compounds are present, for instance the precursor CuSO4 or CuO. A detailed quantitative elemental analysis is needed to determined the product that has been formed.
The reason for the observed increase in TiO2 (001) facets need to be explained. In particular, I do not understand what the authors mean by "It was found that NaOH and (001) TiO2 not only increased the content of (001) facets...".
The bandgap values need to be recalculated. First of all, the UV-Vis spectra do not go to zero at higher wavelengths, which suggest the baseline was not taken properly or the sample scatters a lot of light. This can affect the Tauc plot intercept. Secondly, there is clearly a second transition for the Cu2O/TiO2 composite that can be seen in the Tauc plot, implying that the bandgap should be smaller than 2.64 eV. In fact, literature typically cites a bandgap of 2.0-2.1 eV for Cu2O, which is consistent with the red color of the product. A compound with a 2.64 eV bandgap will look bluish-green in appearance.
Author Response
Comments and Suggestions for Authors
The authors presented a composite Cu2O/TiO2 photocatalyst for ammonia degradation. Major flaws in the experimental design and reporting make it impossible for me to evaluate the performance of the catalyst or the merit of this work. I will need to see a revised version before I can provide a recommendation. Specific issues include:
Typically, photodegradation experiments report the change of the concentration of the pollutant that is being degraded. Here, instead of concentration, a "degradation rate" is reported. No definition of what the authors mean by degradation rate is provided in the manuscript. For instance, what does a degradation rate of 95% mean? Why does the degradation rate drop to 15%? How is the degradation rate calculated? How does the "INNOVA continuous detector" characterize the degradation - is it a gas chromatograph? All these questions need to be answered.
Thank you for your advice. I'm sorry, but we didn't explain the test method of the experiment clearly.The main purpose of this experiment is to degrade ammonia with the prepared composite catalyst Cu2O/TiO2, and the degradation performance of ammonia gas by the loaded photocatalysts under the sunlight simulated by xenon lamp was preliminarily investigated to provide certain theoretical basis for the end treatment of ammonia gas in the field of animal husbandry.
The degradation performance of the photocatalyst was determined by the degradation rate of ammonia. As shown in figure 1 below, there are two test ports before and after the reaction tube for ammonia degradation, namely, the sample inlet and sample outlet, which are connected to INNOVA's inlet and outlet respectively. INNOVA is not gas chromatography, but an instrument for continuously detecting ammonia concentration changes online. The degradation rate at this moment is as follows:
The degradation rate (%) =
Where, The Cin is the ammonia concentration at the inlet; The Cout is the concentration of ammonia after the photocatalytic reaction.The 95% degradation rate means that the degradation of ammonia gas was perfect.
Fig. 1 simple photocatalytic ammonia device
A doubling of gas flow from 0.5L/min to 1L/min caused a dramatic change in the photodegradation profile. This does not make sense and needs to be properly explained. The authors' explanation is inadequate.
Thank you for your advice. In the process of photocatalytic reaction of ammonia, the final reaction is carried out on the surface of photocatalyst, which generally undergoes the following continuous processes: ammonia diffuses to the surface of photocatalytic material; ammonia diffuses from the outer surface to the inner surface of the catalyst; ammonia molecules are adsorbed on the catalyst material; the adsorbed ammonia undergoes photocatalytic reaction; the products produced after photocatalytic reaction are desorbed from the surface of the catalyst; the products diffuse from the surface of the material to the outside surface; and the products desorb from the outside surface of the material to the air.
In this experiment, when other initial conditions were the same and the gas flow rate was gradually increased, the probability of ammonia adsorbed by the material decreases in unit time, the residence time on the catalyst surface was shortened, only NH3 adsorbed on the catalyst surface can be degraded, so the photocatalytic reaction was insufficient, and ultimately the degradation efficiency is reduced. Therefore, in the photocatalytic degradation of ammonia, it is very important to control the appropriate air flow rate.
Although the XRD spectrum of the Cu2O/TiO2 composite contains peaks from both TiO2 and Cu2O, many other peaks can be seen. It is likely that other compounds are present, for instance the precursor CuSO4 or CuO. A detailed quantitative elemental analysis is needed to determined the product that has been formed.
Thank you for your advice. We have carefully checked the other peaks and updated.
The reason for the observed increase in TiO2 (001) facets need to be explained. In particular, I do not understand what the authors mean by "It was found that NaOH and (001) TiO2 not only increased the content of (001) facets...".
Thank you for your advice. Firstly, TEM analysis showed that the composite Cu2O/TiO2 still contained high activity (001) facets. The study titled “Raman Spectroscopy: A New Approach to Measure the Percentage of Anatase TiO2 Exposed (001) Facets, DOI: 10.1021/jp301256h ” that (001) the activity of TiO2 will be greatly enhanced when exposed to (001) facets. For this reason, we have to calculate the (001) facets content of the composite catalyst. We found that the high active surface of the composite was higher than that of the single TiO2. Through the reaction equation of Cu2O, we found that the content of NaOH is excessive. Previous study titled “Photocatalytic Degradation of Methylene Blue over TiO2 Pretreated with Varying Concentrations of NaOH, DOI:10.3390/catal8120575 ” have shown that the treatment of titanium dioxide with NaOH solution is beneficial to the increase of (001) active surface. Therefore, we speculate that excessive NaOH in the reaction system may increase the (001) surface of Cu2O/TiO2 material, which may also be one of the reasons for the increase of Cu2O/TiO2 activity.
The bandgap values need to be recalculated. First of all, the UV-Vis spectra do not go to zero at higher wavelengths, which suggest the baseline was not taken properly or the sample scatters a lot of light. This can affect the Tauc plot intercept. Secondly, there is clearly a second transition for the Cu2O/TiO2 composite that can be seen in the Tauc plot, implying that the bandgap should be smaller than 2.64 eV. In fact, literature typically cites a bandgap of 2.0-2.1 eV for Cu2O, which is consistent with the red color of the product. A compound with a 2.64 eV bandgap will look bluish-green in appearance.
Thank you for your advice. Another reviewer gave us the calculation method. We re-measured the UV-Vis diffuse reflectance of the catalyst and added the Cu2O spectrum. Finally, the band gap was recalculated according to the method used in the literature.

Reviewer 2 Report
The authors developed heterostructures of Cu2O/TiO2 by impregnation-reduction. They showed that the crystal has a highly active facet content. The authors investigated the photocatalytic degradation of ammonia using this material showing that its is more efficient than P25, Cu2O, and TiO2.
It is a strong paper and I support its publications
I have the following minor comments:
1- Incorporate some of the mots recent relevant reports on catalytic oxides including: Journal of Materials Chemistry A 5, 24223, 2017 and Advanced Functional Materials 27, 1702295, 2017 and Journal of Materials Chemistry C 2, 5247, 2014
2- You need to identify the extra peaks which are seen in the XRD of the Cu2O/TiO2
3- The bandgaps are incorrect - the measurement should be crosses against the tangents to the graphs and not the x axis
4- There is a diffusion of the two Cu2O and TiO2 and as such there are perhaps binary oxide formation - so Fig 10 b is not a complete depiction of the photocatalysis
Author Response
Comments and Suggestions for Authors
The authors developed heterostructures of Cu2O/TiO2 by impregnation-reduction. They showed that the crystal has a highly active facet content. The authors investigated the photocatalytic degradation of ammonia using this material showing that its is more efficient than P25, Cu2O, and TiO2.
It is a strong paper and I support its publications
I have the following minor comments:
1- Incorporate some of the mots recent relevant reports on catalytic oxides including: Journal of Materials Chemistry A 5, 24223, 2017 and Advanced Functional Materials 27, 1702295, 2017 and Journal of Materials Chemistry C 2, 5247, 2014
Thank you for letting us know interesting materials like MoO3 and Ga2O3, and the applications of copper series catalysts in other fields. Therefore, we introduced and cited these literatures in this article.
2- You need to identify the extra peaks which are seen in the XRD of the Cu2O/TiO2
Thank you for your suggestion. We have carefully checked the other peaks and updated.
3- The bandgaps are incorrect - the measurement should be crosses against the tangents to the graphs and not the x axis
Thank you for your suggestion. Another reviewer gave us the calculation method. We re-measured the UV-Vis diffuse reflectance of the catalyst and added the Cu2O spectrum. Finally, the band gap was recalculated according to the method used in the literature.
4- There is a diffusion of the two Cu2O and TiO2 and as such there are perhaps binary oxide formation - so Fig 10 b is not a complete depiction of the photocatalysis
Thank you for your suggestion. We have improved our explanation.

Reviewer 3 Report
In this manuscript a heterogeneous composite catalyst Cu2O/(001) TiO2
was prepared, characterized and applied in the photocatalytic ammonia
degradation of ammonia.
The research is interesting but any question should be clarified:
In the UV-Vis absorption analysis of catalysts the calculation of band gap should be done as reported in Applied Surface Science 441 (2018) 575–587 with the addition of relative reference.
For the better results interpretation, I suggest the calculation of degradation rates in the chapter "Degradation of Ammonia by the Catalyst" and the collection of the related results in a table. A discussion of results should be presented.
The term Eg is present in Raman spectra as Peak at 144 cm-1 and also as the band gap. This creates confusion and must be explained differently.
The results of chapter "Degradation of Ammonia by Polyester Fiber Carrier" are not clear: explain better.
I reccomended the authors to insert all preparation methods in the chaper 2. Materials and Methods and to correct any writing errors.
Author Response
Comments and Suggestions for Authors
In this manuscript a heterogeneous composite catalyst Cu2O/(001) TiO2 was prepared, characterized and applied in the photocatalytic ammonia degradation of ammonia.
The research is interesting but any question should be clarified:
In the UV-Vis absorption analysis of catalysts the calculation of band gap should be done as reported in Applied Surface Science 441 (2018) 575–587 with the addition of relative reference.
Thank you for your advice. The UV-Vis DRS of the catalysts were measured again, and the Cu2O spectrum was added, and the calculation method of this document is also consulted.
For the better results interpretation, I suggest the calculation of degradation rates in the chapter "Degradation of Ammonia by the Catalyst" and the collection of the related results in a table. A discussion of results should be presented.
Thank you for your advice. But I don't think it's possible, because INNOVA is an on-line gas concentration monitoring instrument, so each set of data has many values, which is very verbose in the article. but I still attached the data(Word).
The term Eg is present in Raman spectra as Peak at 144 cm-1 and also as the band gap. This creates confusion and must be explained differently.
Thank you for your advice. Firstly, TEM analysis showed that the composite Cu2O/TiO2 still contained high activity (001) facets. For this reason, we have to calculate the (001) facets content of the composite catalyst. The literature have reported the calculation method of (001) surface content, for example, the title is “Anatase TiO2 Nanoparticles with Exposed {001} Facets for Efficient Dye-Sensitized Solar Cells” the DOI: 10.1038/srep12143. Meanwhile, for your convenience, I have screenshots below. The percentage of {001} facets was calculated as 34% by measuring the peak intensity ratio of the Eg (at 144 cm−1) and A1g (at 514 cm−1) peaks. Although Eg is present in Raman spectra as Peak at 144 cm-1 and also as the band gap, and I don't think it will affect the calculation of (001) content.
Figure. (001) facets Content Calculation
The results of chapter "Degradation of Ammonia by Polyester Fiber Carrier" are not clear: explain better.
Thank you for your advice. We have improved our explanation
I reccomended the authors to insert all preparation methods in the chaper 2. Materials and Methods and to correct any writing errors.
Thank you for your advice. We have changed all preparation methods.

Round 2
Reviewer 1 Report
The authors have addressed my concerns, and I am willing to recommend the manuscript for publication.
Reviewer 3 Report
With these corrections I suggest the publication.